# Cell Type-Specific Induction of Inflammation-Associated Genes in Crohn’s Disease and Colorectal Cancer

**DOI:** 10.3390/ijms23063082

**Published:** 2022-03-12

**Authors:** Dominik Saul, Luísa Leite Barros, Alexander Q. Wixom, Benjamin Gellhaus, Hunter R. Gibbons, William A. Faubion, Robyn Laura Kosinsky

**Affiliations:** 1Division of Endocrinology, Mayo Clinic, Rochester, MN 55905, USA; 2Robert and Arlene Kogod Center on Aging, Mayo Clinic, Rochester, MN 55905, USA; 3Department of Trauma, Orthopedics and Reconstructive Surgery, Georg-August-University of Göttingen, 37075 Göttingen, Germany; benjamin.gellhaus@stud.uni-goettingen.de; 4Division of Gastroenterology and Hepatology, Mayo Clinic, Rochester, MN 55905, USA; barros.luisa@mayo.edu (L.L.B.); wixom.alexander@mayo.edu (A.Q.W.); gibbons.hunter@mayo.edu (H.R.G.); faubion.william@mayo.edu (W.A.F.); 5Department of Gastroenterology, School of Medicine, University of São Paulo, São Paulo 05403-000, Brazil

**Keywords:** inflammatory bowel disease, single cell sequencing, gene expression, transcriptomics, next generation sequencing, Crohn’s disease, colorectal cancer, ulcerative colitis

## Abstract

Based on the rapid increase in incidence of inflammatory bowel disease (IBD), the identification of susceptibility genes and cell populations contributing to this condition is essential. Previous studies suggested multiple genes associated with the susceptibility of IBD; however, due to the analysis of whole-tissue samples, the contribution of individual cell populations remains widely unresolved. Single-cell RNA sequencing (scRNA-seq) provides the opportunity to identify underlying cellular populations. We determined the enrichment of Crohn’s disease (CD)-induced genes in a publicly available Crohn’s disease scRNA-seq dataset and detected the strongest induction of these genes in innate lymphoid cells (ILC1), highly activated T cells and dendritic cells, pericytes and activated fibroblasts, as well as epithelial cells. Notably, these genes were highly enriched in IBD-associated neoplasia, as well as sporadic colorectal cancer (CRC). Indeed, the same six cell populations displayed an upregulation of CD-induced genes in a CRC scRNA-seq dataset. Finally, after integrating and harmonizing the CD and CRC scRNA-seq data, we demonstrated that these six cell types display a gradual increase in gene expression levels from a healthy state to an inflammatory and tumorous state. Together, we identified cell populations that specifically upregulate CD-induced genes in CD and CRC patients and could, therefore, contribute to inflammation-associated tumor development.

## 1. Introduction

With approximately 1.6 million individuals affected and 70,000 new diagnoses each year in the US alone, inflammatory bowel disease (IBD), including Crohn’s disease (CD) and ulcerative colitis (UC), represents a major health burden [1]. Nearly 250 single nucleotide polymorphism (SNP)-tagged loci [2,3,4,5] and chromosomal alterations [6,7], as well as splicing variants [8,9], were linked to an elevated risk of developing IBD. In addition, several studies investigated IBD-induced gene expression changes in the past, including expression quantitative trait loci (eQTL) approaches [10]. However, due to the analysis of whole-tissue samples, the contribution of individual cell populations to the expression of IBD risk genes remains to be elucidated.

Researchers have identified several cell types that promote intestinal inflammation; for instance, via the secretion of pro-inflammatory cytokines. As summarized by Silva et al., macrophages and dendritic, epithelial and T cells, i.e., Th1, Th2, Th17 and regulatory T cells, contribute to intestinal inflammation by cytokine production [11]. Besides the aberrant secretion of pro-inflammatory cytokines, these disruptions in epithelial cell function and barrier integrity are major events in IBD. In a recent single-cell profiling study, Parikh et al. determined the contributions of a distinct epithelial cell subpopulation to the barrier disruption in IBD [12]. 

Since most polymorphisms linked to IBD are as of yet unannotated, and eQTL studies to date are tissue-based, this work aimed to link previous SNP- and gene expression-associated knowledge to IBD biology. In addition, in the current study, we sought to determine to what extent individual cell populations contribute to the upregulation of CD risk genes detected in whole-tissue specimens in earlier studies. Finally, we aimed to verify whether a subsequent transformation to a carcinomatous state can be mirrored on a transcriptome-wide and single-cell level. We hypothesized that only specific cell types mediate the induction of these genes. Specifically, due to their key role in maintaining the intestinal barrier function and their presumably high abundance in samples utilized for bulk sequencing datasets, we expect that epithelial cells largely contribute to the induction of CD-risk genes. In addition, we hypothesize that distinct immune regulatory cells play major roles in mediating inflammatory signaling. An in-depth characterization of CD-induced genes and the availability of next-generation sequencing data on a single cell level allow for the verification of CD risk genes and identify causative cell populations and their relevance in the context of local cellular heterogeneity. We confirmed the upregulation of previously suggested CD risk genes in inflamed terminal ileum on a single cell level. Six cell populations were identified as the major drivers of the upregulation of these genes in CD. Intriguingly, this gene signature was highly enriched in UC-associated neoplasia, as well as sporadic colorectal cancer (CRC). Indeed, these six cell types displayed a gradual increase in gene expression levels when comparing the control to inflamed and tumorous tissue.

## 2. Results

### 2.1. Only Distinct Cell Populations Contribute to the Upregulation of Inflammation-Induced Genes in CD Lesions

Genome-wide association studies (GWAS) have identified several SNPs to be linked with an increased susceptibility of developing IBD. In a comprehensive study by Peloquin et al., the expression of CD risk genes was evaluated by comparing inflamed mucosal intestinal tissues from IBD patients to healthy controls [10]. Initially, to underscore the clear distinction between SNPs and gene expression data, we compared these genes to previously identified loci associated with SNPs [13,14]. Intriguingly, only 31% of genes upregulated in chronic intestinal inflammation and 19% of downregulated genes were suggested to be associated with an increased risk towards developing IBD in SNP-based studies (Figure 1A). Thus, genes differentially expressed in CD were associated with loci beyond the recognized SNPs. Gene ontology confirmed that SNP-containing loci, as well as genes upregulated in CD, are related to cytokine production, cytokine-mediated signaling pathways and an inflammatory response (Figure 1B). In contrast, these processes were not affected by factors downregulated in CD, emphasizing the regulatory relevance of genes induced during inflammatory processes. Since previous RNA sequencing approaches and the subsequent identification of risk genes was based on bulk analyses, the contribution of distinct cell types remains unknown. Therefore, we aimed to assess which cell populations display an upregulation of genes suggested by the work of Peloquin et al. Initial gene-ontology-based analyses using the PanglaoDB database predicted that this gene set is enriched in immune regulatory cells, including macrophages, basophils, neutrophils, dendritic cells and T cells (Figure 1C). Next, an in-depth analysis of the differential expression of these CD-induced genes in distinct cell types was performed using publicly available scRNA-seq data of CD lesions and uninflamed control tissue [15]. After clustering all cells, as suggested by the original study (Appendix A), we enriched every cell type for the generated gene set (Figure 1D). In pairwise comparisons, we discovered that innate lymphoid cells (ILC1), highly activated T cells and dendritic cells (DCs), pericytes and activated fibroblasts, as well as epithelial cells, displayed the highest upregulation of CD-induced genes. Together, while previously published gene expression analyses largely contributed to our knowledge about CD risk genes, the integration of scRNA-seq data allows for the identification of cellular key players.

### 2.2. Cell Type-Specific Enrichment and Co-Expression Patterns of CD-Induced Genes 

Based on high expression levels of CD-induced genes, we focused on ILC1, highly activated T cells and dendritic cells, pericytes and activated fibroblasts, as well as epithelial cells, in our subsequent analyses. We confirmed the enrichment of CD-induced genes in the respective cell types isolated from inflamed lesions by gene set enrichment analysis (GSEA; Figure 2A). Similarities between cells isolated from NAT and CD were evaluated using uniform manifold approximation and projection (UMAP) plots (Figure 2B). Generally, transcriptomes of both conditions appeared as similar in most cell types. Notably, activated fibroblasts isolated from CD lesions were highly distinct from control cells. To further elucidate how the respective cell populations contribute to the induction of CD risk genes in chronically inflamed intestinal tissue, we defined the most upregulated susceptibility genes per cell type (Figure 2C). Within the ILC1 cluster, HLA-C, TNFAIP3 and LITAF showed the most profound induction in inflamed tissue, whereas GPR183, TNFAIP3 and PLEK were highly upregulated in activated DCs, OLFML3, CXCL2 and CXCL3 in activated fibroblasts, HLA-C, ADA and IRF1 in highly activated T cells, CCL2, ERRFI1 and CXCL2 in pericytes and HLA-C, PIGR and ERRFI1 in epithelial cells. A correlation plot of all CD risk genes revealed the presence of a gene subset with a high co-expressional pattern (Appendix A). Therefore, we generated correlation plots displaying the most relevant co-expression patterns per cell type, confirming a profound pairwise interconnectivity of distinct CD-induced genes (Figure 2D).

### 2.3. CD-Induced Genes Are Upregulated in UC-Associated Neoplasia and Sporadic CRC

While the probability of developing colorectal cancer (CRC) is higher in UC than CD patients [18], the overall risk of patients with colonic IBD developing CRC is increased 1.7-fold [19]. Interestingly, the recent gene-expression-based identification of two clinically relevant CD subtypes, colon- and ileum-like CD [20], suggests that IBD subclasses are not always clearly distinct on a molecular level. Indeed, when comparing genes induced in UC and CD, a substantial overlap was detected (Figure 3A). Thus, we assessed the expression of CD-induced genes in a microarray dataset analyzing tissue isolated from healthy controls, UC lesions and UC-associated neoplasia (UCneo [21]). Indeed, a significant enrichment of CD-induced genes was detected in UCneo samples (Figure 3B,C). To extend this finding, we utilized two independent bulk RNA-seq datasets comparing CRC to normal control tissue [22,23]. Notably, GSEA demonstrated that there is a significant enrichment of CD-induced genes in sporadic colorectal cancer (Figure 3D). Using publicly available TCGA data, we evaluated the effect of a high expression of CD-induced genes on the overall survival of CRC patients (Figure 3E). In fact, high-level genes, particularly highly expressed in UCneo, i.e., CXCL6, ITGAM, TDO2, CXCL1, RGS1, FCGR3A, CXCL8 and FCGR3B, displayed a heterogeneous effect on patient survival. This finding is in agreement with a recent study by Vitali et al., who detected no difference between the outcome of UC-associated and sporadic CRC [24]. In summary, CD-induced genes are upregulated in UC-associated neoplasia, as well as sporadic CRC, supporting the established concept of chronic inflammation as a predisposing factor for intestinal tumorigenesis.

### 2.4. Enrichment of CD Risk Genes in Sporadic CRC on a Single-Cell Level

We aimed to elucidate the relevance of CD risk genes in CRC in more detail and to acquire knowledge on cell type-specific functions. Subsequently, we analyzed a scRNA-seq dataset containing CRC and normal control samples [26]. To ensure the comparability between the CD and CRC in scRNA-seq data, the same cell type annotation suggested in the CD study was applied to the CRC dataset. After assigning the cell types within this study (Figure 4A, Appendix A), we evaluated the enrichment of CD risk genes in CRC and control cells. Besides an upregulation in stroma cells, we detected a significant enrichment of CD-induced genes in ILC1, highly activated T cells and dendritic cells, pericytes and activated fibroblasts, as well as epithelial cells (Figure 4B). Therefore, we decided to characterize these cell populations in more detail.

### 2.5. Enrichment and Co-Expression Patterns of CD-Induced Genes in Distinct Cell Populations in CRC 

Notably, when determining the enrichment of CD risk genes in these cell types in CRC using GSEA, only ILC1, activated DCs, fibroblasts and highly activated T cells reached statistical significance. While only a few genes were enriched in ILC1, pericytes and epithelial cells showed only a positive trend towards the induction of CD risk genes in CRC (*p* = 0.051 and *p* = 0.0599, respectively; Figure 5A). As suggested by UMAPs, pericytes and epithelial cells isolated from tumors were highly distinct from control cells (Figure 5B). Next, we determined the most upregulated CD risk genes per cell population. Within the ILC1 cluster, *RGS1*, *STAT3* and *TNFRSF18* showed the most significant induction in CRC tissue, whereas *RGS1*, *FCGR2A* and *CXCL3* were highly upregulated in activated DCs, *NUPR1*, *CXCL2* and *ZFP36L2* in activated fibroblasts, *RGS1*, *GPR183* and *TNFAIP3* in highly activated T cells, *XBP1*, *TNFAIP3* and *CXCL2* in pericytes and *CXCL1*, *CXCL2* and *CEBPB* in epithelial cells (Figure 5C). While the top three upregulated CD risk genes were mostly different among cell types when comparing CD and CRC, the top 10 candidates showed several overlaps. Similar to our previous analyses, correlation plots displaying the most relevant co-expression patterns per cell type suggested a profound pairwise interconnectivity of distinct genes (Figure 5D, Appendix A).

### 2.6. The Expression of CD Risk Genes Gradually Increases from Control NAT, CD to CRC Cells

Finally, in order to evaluate whether CD risk genes display a gradual upregulation from control to CD and, subsequently, transformed cells, we integrated and harmonized the CD and CRC scRNA-seq dataset. Determining the enrichment of CD-induced genes revealed a gradual and significant increase in all six previously identified cell populations, i.e., ILC1, highly activated T cells and dendritic cells, pericytes and activated fibroblasts, as well as epithelial cells (Figure 6). As a control, this phenomenon was tested in cell populations that did not display a significant induction of CD risk genes in CD or CRC earlier. As exemplarily shown for the naïve T cells, this gradual increase was not detected (Appendix A).

## 3. Discussion

While several studies identified IBD susceptibility genes, as well as (partially unannotated) risk loci, there is a lack of a global evaluation of the cellular contribution on a single-cell level. In this study, we provide evidence for the cellular composition underlying the transcriptional enrichment of risk genes in CD and CRC patients. 

Publicly available scRNA-seq data of inflamed terminal ileum specimens and uninvolved controls revealed a profound upregulation of CD-induced genes in ILC1, highly activated T cells and dendritic cells, pericytes and activated fibroblasts, as well as epithelial cells. Interestingly, UC-associated neoplastic lesions displayed an enrichment in these genes as well. Therefore, we sought to investigate the extent to which this signature plays a role in sporadic CRC. Initial GSEA approaches revealed a highly significant upregulation of CD-induced genes in two sporadic CRC bulk RNA-seq datasets. Indeed, when extending our approach to a single-cell level, we discovered an upregulation of these genes in the aforementioned cell types in CRC. Finally, to further elucidate this inter-relation, we integrated and harmonized the CD and CRC scRNA-seq datasets, demonstrating a gradual increase from control to CD and CRC in these cell populations.

In agreement with our findings, these cell types have been linked to intestinal inflammation and tumorigenesis in earlier studies. 

The ILC1 population was first described by Bernink et al., and their importance in the inflamed intestine of Crohn’s patients, especially via INF-γ and the pathogenesis of gut mucosal inflammation, has been emphasized [27]. Interestingly, the high *INFG* expression levels that we detected in CRC samples compared to controls is in support of these data. ILC1 innate lymphoid cells were demonstrated to hold a crucial role in the IL22 production and, thus, regulation of inflammatory bowel disease, but are markedly distinct from conventional natural killer (NK) cells [28,29]. Interestingly, CRC progression was recently suggested to be mediated by distinct ILC populations, while the presence of ILC1-like cells was identified in CRC lesions [30].

The relevance of activated T cells in IBD is widely characterized [31], and their multifaceted regulatory activity in IBD has been previously demonstrated by our group [32,33]. Their pro-inflammatory function, including the upregulation of CXCL1-3 and IFNG, was described in order to contribute to the development of CRC [34]. Similarly, activated DCs accumulate in the colon of UC patients [35] and their suppression by cancer cells has been hypothesized as a mechanism to avoid immune surveillance [36]. Notably, an increased abundance of activated DCs in invasive tumors, along with lymph node invasion, was observed [37].

Among these immune cell populations (ILC1, activated T cells and activated DCs), a recurring expressional pattern of TNFα-signaling-related genes was observed. For instance, *TNFAIP3* in ILC1 and activated DC is induced by TNFα, as are IRF1 (and related interferon-pathway genes) in all of these cell types and *STAT1*, leading to an autocrine loop of inflammation [38]. Similarly, the induction of HLA-C, which we have shown in all of these populations, can be explained via induction by TNFα [39]. The importance of these expression patterns is highlighted by the application of TNFα inhibitors as a mainstay therapy for over 20 years [40,41,42].

Pericytes, residing at the interface between the endothelium and interstitium, mostly contribute to inflammation-associated fibrosis, whereas their role in IBD [43], as well as CRC, is not fully understood. Interestingly, earlier studies suggest a functional relevance in the tumor microenvironment and a potential involvement in antiangiogenic therapies [44]. Activated fibroblasts were just recently discovered to be critical pro-inflammatory players in both CD and the development of CRC. These cells mediate increased levels of CXCL1, -2 and -3 and further pro-inflammatory cytokines and, therefore, were suggested to drive pro-tumorigenic development in IBD [45]. Epithelial cells are the main cell type described as being involved in the development of colorectal cancer [46,47]. While their heterogeneity and importance in barrier breakdown in IBD was highlighted in the past [12], our findings support their key role in IBD and CRC [48].

Among some ILC1, activated DCs, pericytes and activated fibroblasts, the levels of the intercellular adhesion molecule 1 (ICAM1) increased in the diseased state. Intriguingly, ICAM1 was increased in the serum of Crohn’s patients and declined significantly during treatment and after remission [49]. Notably, we identified a potential cellular origin of this potentially impactful serum marker.

The analysis of CD and UC data within the same study may have reduced the pathogenic insights into the individual respective disease. However, the large proportion of CD-induced genes upregulated in UC suggests a general deregulation of distinct genes, as well as signaling pathways, in IBD. Future research will aim to specifically verify and expand our findings in CD and UC in separate studies. 

## 4. Materials and Methods

### 4.1. CD Signature

Genes differentially expressed in CD patients were obtained from an earlier study by Peloquin et al. Initially, we selected all genes upregulated (*n* = 176) or downregulated (*n* = 91) in CD [10]. As explained in the results section, we later focused on CD-induced genes (also referred to as CD risk genes throughout this manuscript). This signature was enriched in single-cell sequencing datasets using the enrichIT function within the escape package, in groups of 1000 cells (v. 1.3.3, [50]). Out of these 176 genes, 153 were detected in both single-cell sequencing datasets. Therefore, heatmap analyses and pairwise comparisons focus on these 153 genes (Appendix A). To compare genes upregulated in CD and UC, UC-induced genes were obtained from the same study by Peloquin et al. [10].

### 4.2. Analysis of Microarray Sequencing Data

To identify genes associated with UC, as well as UC-associated neoplasia, publicly available microarray data (healthy controls, *n* = 5; quiescent UC, *n* = 3, UC and neoplasia, *n* = 10) were analyzed ([21], GSE37283) using Transcriptome Analysis Console Tac Software (TAC4.0) (appliedbiosystems, Thermo Fisher, Waltham, MA, USA) with the Array Type HT_HG-U133_Plus_PM.

### 4.3. Analysis of CRC Bulk mRNA Sequencing Data

The enrichment of CD risk genes in CRC specimens was tested in two publicly available bulk mRNA datasets. In the study by Paredes et al. ([22], GSE146009, Caucasian Americans) 18 tumor and 17 adjacent non-transformed tissues were compared, while Kim et al. ([23], GSE50760) assessed transcriptome-wide changes in 18 primary CRC and 18 normal colon samples. These bulk mRNA-seq data were analyzed (GSE146009, GSE50760) as described earlier ([51]). Briefly, raw counts were converted into a matrix, before DESeq2 (1.34.0) was used. Determination of differentially expressed genes (DEGs) was performed using DESeq2 (lfcThreshold = 0, alpha = 0.1, min. count = 0.5). An exemplarily RNA-seq analysis vignette is provided as R notebook in our previous study [51]. Gene set enrichment analysis (GSEA, v. 4.2.2, Broad Institute, Inc., Massachusetts Institute of Technology, and Regents of the University of California, Massachusetts, CA, USA) was performed with default settings (1000 permutations for gene sets, Signal2Noise metric for ranking genes).

### 4.4. Single-Cell RNA-seq (scRNA-seq) Analysis

Differential expression of CD risk genes in distinct cell populations was determined by analyzing publicly available single-cell sequencing datasets comparing CD to uninflamed intestinal tissue (GSE134809; CD, *n* = 11; uninflamed, *n* = 11, [15]) and CRC specimens to matched normal mucosa (E-MTAB-8410; CRC, *n* = 23; normal, *n* = 10; [26]). Cells with at least 500 unique molecular identifiers (UMIs), log10 genes per UMI ˃0.8 and ˃250 genes per cell and a mitochondrial ratio of less than 20% were extracted, normalized and integrated using the Seurat package (v 4.0.6) in R 4.0.3, as demonstrated in our previous studies ([51,52]). The cell annotation utilized in the CD study was provided by the authors, and uninflamed, normal adjacent tissue (NAT) was referred to as “NAT”, while involved ileal samples have been named “IBD” ([15], GSE134809). Detailed information on sample characteristics, conditions and cell numbers per cluster from the CD scRNA-seq dataset are summarized in Appendix A. To determine common mechanisms in CD and CRC samples, the same cell type annotation suggested in the CD study was selected for the CRC data. The sample classification into “tumor” and “normal” was provided by the authors [26]. Sample characteristics are summarized in Appendix A.

The normalization, scaling and clustering followed the recommendations of the Seurat package [53]. Comparisons between IBD and NAT, as well as normal and tumor samples, were performed using the “FindMarkers” function (Seurat package) and the “MAST” package (1.16.0, [54], logfc.threshold = 0, test.use = ”MAST”, only.pos = FALSE, min.pct = 0.0). For downstream analyses, ggplot2 (3.3.5) was utilized. For pairwise comparisons, a non-parametric Wilcoxon signed-rank test was applied (ggpubr 0.4.0). GSEA analysis was conducted using clusterProfiler (3.18.1). Correlation plots were designed with corrplot (0.92), ggsci (2.9) and ggExtra (0.9) and heatmaps using the DEP package (1.1.5).

The two scRNA-seq datasets were harmonized with the harmony package (0.1.0) according to the authors instructions [55]. Briefly, the separate Seurat objects were merged with the Seurat package, normalized and scaled before the variation of gene expression was embedded in a PCA, normalized for library size. The eigenvalue-scaled eigenvectors were used as harmony input before the batch variables (NAT, IBD and tumor) were used to integrate, which returned a Seurat object with harmony correction.

Correlation analyses were performed according to the suggestions of Iacono et al. [56] using the bigSCale framework (2.0). Briefly, modules for differential expression were generated in a newly created regulatory network. After clustering based on granularity, iterative differential expression analysis between cluster pairs was performed. In the thereby created Z-score space, a solid measure of linear correlation via Spearman coefficients was used to correlate gene pairs.

### 4.5. Statistics and Graphs

Statistical analyses were performed using a D’Agostino and Pearson test for normality. If passed, an unpaired *t*-test was performed. Otherwise, a Mann–Whitney test was performed (* *p* ≤ 0.05, ** *p* ≤ 0.01, *** *p* ≤ 0.001). Correlation analyses were performed with Spearman’s correlation. Raw count comparisons were calculated with a one-way ANOVA.

Graphs were designed using GraphPad Prism 9.2.0 (GraphPad Software, Inc., San Diego, CA, USA), BioRender.com and R (4.0.3).

## 5. Conclusions

We are the first group to provide evidence for the relevance of these distinct cell types in driving CD-associated gene expression patterns. Therefore, these cells are likely to contribute to the development of chronic intestinal inflammation and progression into CRC. Further studies will validate these findings in primary cells and will help to unravel the role of these cells in the local inflammatory environment, as well as their therapeutic vulnerability towards existing and novel treatments.

## Figures and Tables

**Figure 1 ijms-23-03082-f001:**
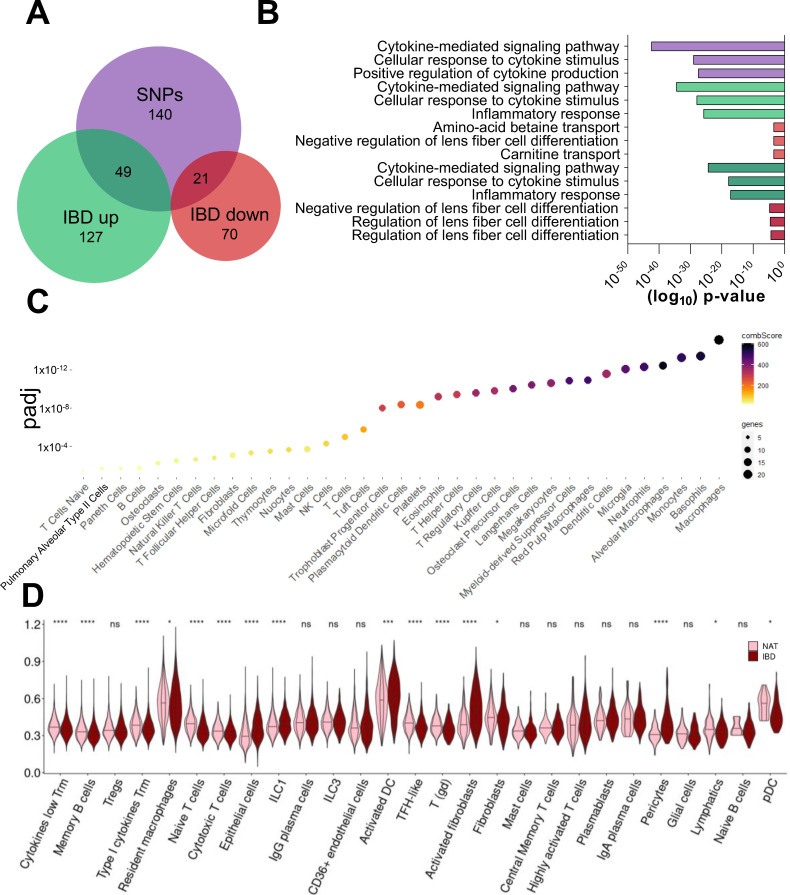
Identification of cell populations upregulating CD risk genes. (**A**) Venn diagram displaying SNP-containing loci linked to IBD [13,14] with genes up- or downregulated in CD [10]. (**B**) Gene ontology (GO Biological Process, EnrichR [16]) revealed that SNP-containing loci (purple), genes upregulated in CD (green) and their overlap (teal) are related to inflammatory pathways, while genes downregulated in CD (red) and their overlap with SNP-containing regions (dark red) are not. (**C**) Dot plot depicting cell types related to CD-induced genes according to PanglaoDB Augmented 2021 [17]. The dot color reflects a combination of *p*-value and z-score, referred to as combined score, whereas the dot size correlates to the number of enriched genes. (**D**) Violin plot indicating the pairwise enrichment of CD-induced genes in scRNA-seq data from CD lesions (IBD; dark red) and uninflamed control tissue (NAT; light pink; [15], Wilcoxon-test). * *p* < 0.05, *** *p* < 0.001, **** *p* < 0.0001.

**Figure 2 ijms-23-03082-f002:**
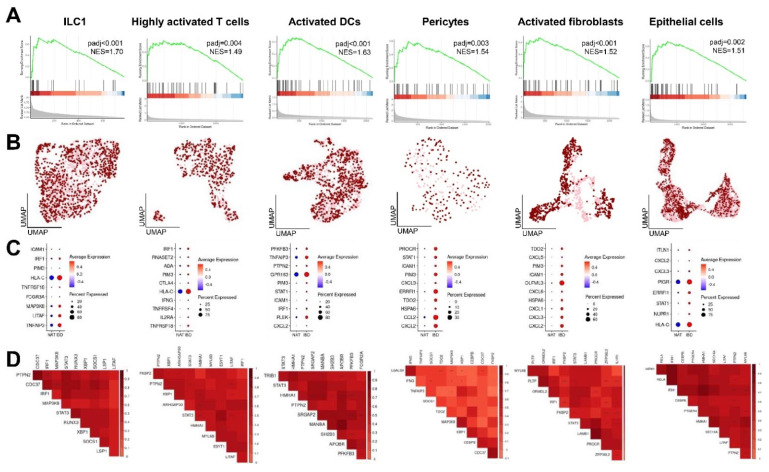
Cell type-specific upregulation and co-expression of risk genes in CD lesions. (**A**) GSEA confirms the enrichment of CD-induced genes in ILC1, highly activated T cells and dendritic cells, pericytes and activated fibroblasts, as well as epithelial cells, isolated from inflamed CD lesions. (**B**) UMAPs displaying similarities between control (light pink) and inflamed (dark red) cells based on dimensional reduction. While gene expression patterns among those conditions appear comparable, inflamed activated fibroblasts are highly distinct from controls. (**C**) Dot plots displaying ten genes per cell type with the most significant differential expression. Dot size indicates the percentage of cells expressing the respective genes, dot color represents expression intensity. (**D**) Correlation plots revealing co-expression patterns of CD-induced genes.

**Figure 3 ijms-23-03082-f003:**
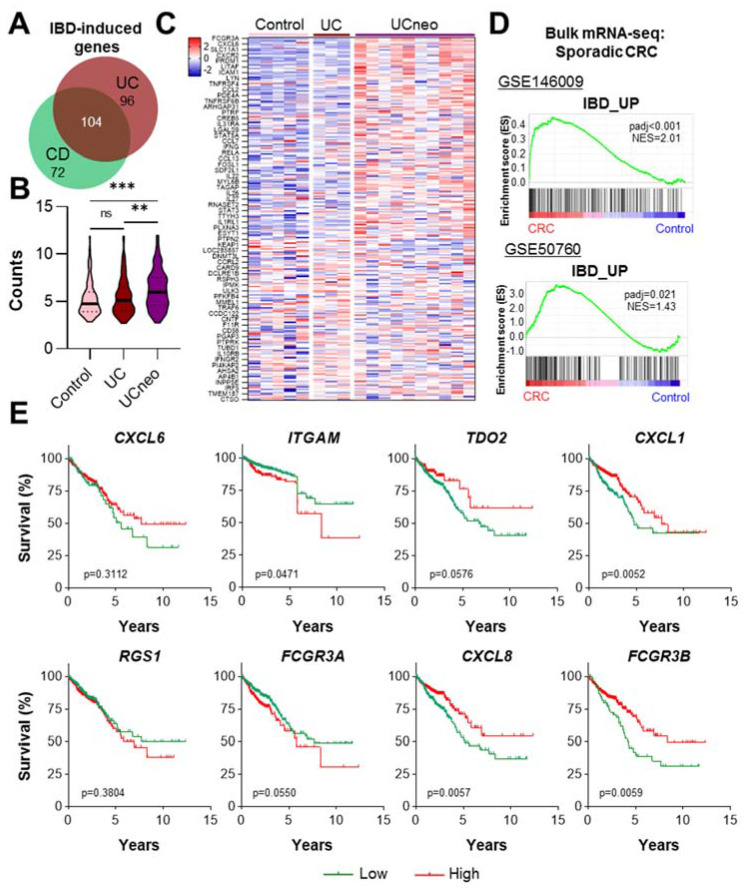
Enrichment of CD risk genes in UC-associated neoplasia and sporadic CRC. (**A**) Genes upregulated in UC and CD regions were obtained from a study by Peloquin et al. [10] and overlapped. (**B**) Microarray analysis revealed only a slight upregulation of CD risk genes in UC patients (*n* = 3) compared to controls (*n* = 5), whereas a profound induction was observed in UC patients with neoplasia (*n* = 10; [25], one-way ANOVA). (**C**) The corresponding heatmap displays the expression of all CD risk genes among control, UC and UCneo patients [25]. (**D**) The enrichment of CD-induced genes in sporadic CRC was verified in two bulk mRNA-seq datasets comparing normal tissue to tumor tissue using GSEA [22,23]. CD risk genes displayed a significant upregulation in sporadic CRC compared to control tissues (*p* < 0.001, NES = 2.01 and *p* = 0.021, NES = 1.43, respectively). (**E**) Survival curves of CRC patients with low (green) or high (red) expression of CD-induced genes significantly upregulated in UCneo samples. ** *p* < 0.01, *** *p* < 0.001.

**Figure 4 ijms-23-03082-f004:**
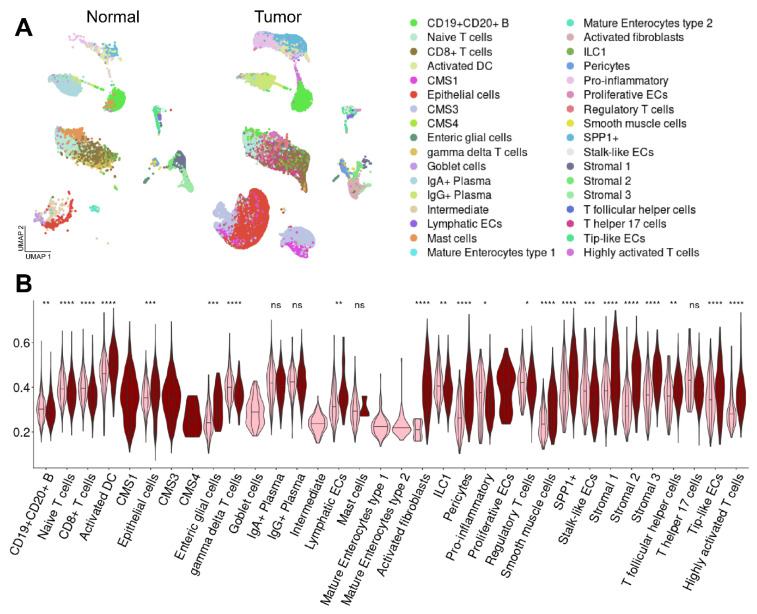
Enrichment of CD risk genes in UC-associated neoplasia and sporadic CRC. (**A**) Genes upregulated in UC and CD regions were obtained from a study by Peloquin et al. [10] and overlapped. (**B**) Microarray analysis revealed only a slight upregulation of CD risk genes in UC. * *p* < 0.05, ** *p* < 0.01, *** *p* < 0.001, **** *p* < 0.0001.

**Figure 5 ijms-23-03082-f005:**
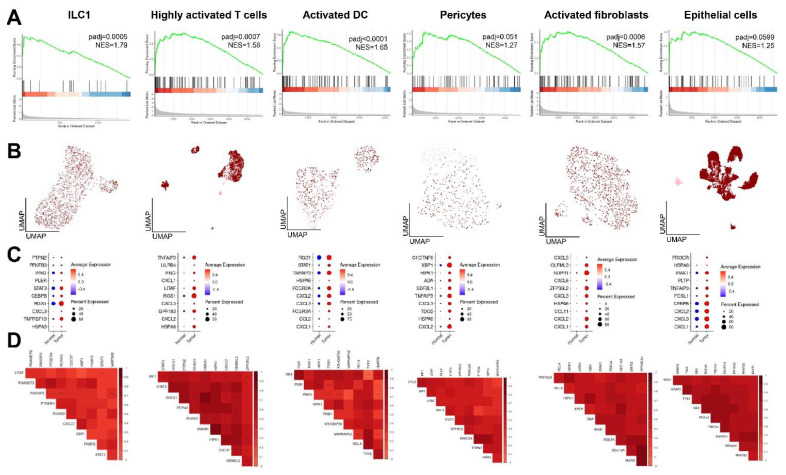
Cell-population-specific induction and co-expression of CD risk genes in sporadic CRC. (**A**) GSEA reveals the statistically significant enrichment of CD-induced genes in ILC1, activated DCs, activated fibroblasts and highly activated T cells isolated from colorectal tumors. (**B**) UMAPs displaying gene expression-based similarities between control (light pink) and transformed (dark red) cells. Pericytes and epithelial cells isolated from tumors appear highly distinct from controls. (**C**) Dot plots depicting genes with the most significant differential expression per cell population. Dot size indicates the percentage of cells expressing the respective genes, dot color represents expression intensity. (**D**) Correlation plots revealing co-expression patterns of CD-induced genes in sporadic CRC.

**Figure 6 ijms-23-03082-f006:**
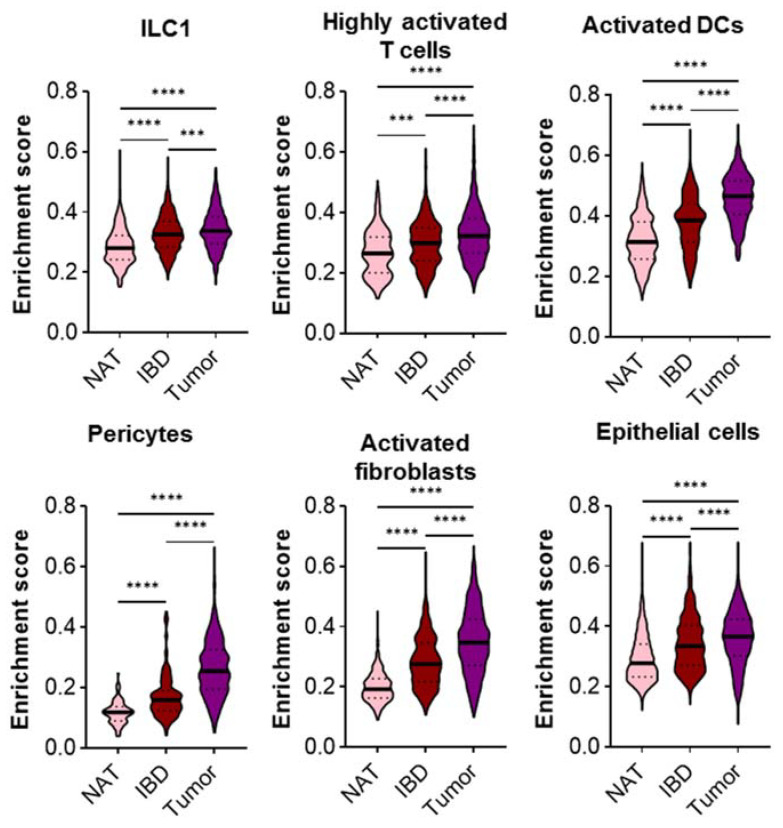
Enrichment of CD risk genes along control, inflammatory and finally tumorous state. After integration and harmonization, the CD and CRC scRNA-seq datasets [15,26] were normalized and scaled together, and an enrichment of CD risk genes was performed. ILC1, highly activated T cells and dendritic cells, pericytes and activated fibroblasts, as well as epithelial cells, displayed a significant upregulation during inflammation, as well as CRC. Kruskal–Wallis test with correction for multiple comparisons by Dunn’s test, α = 0.05. *** *p* < 0.001, **** *p* < 0.0001.

## Data Availability

Data sharing not applicable.

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
