# Peer review of "Cell Type-Specific Induction of Inflammation-Associated Genes in Crohn’s Disease and Colorectal Cancer"

_ijms, 2022, doi:10.3390/ijms23063082_

Round 1

Reviewer 1 Report

The manuscript by Saul et al. take advantage of publically available single-cell RNAseq data sets to identify the cell types displaying changes in Crohn's disease-associated gene signatures. They further assess these cell-specific gene changes in additional UC-associated neoplasia and sporadic colon cancer datasets in order to track their invovlement during disease progression. The authors also nicely summarize the known and uknown functions for each of the described cell types during IBD and CRC in the discussion. It would be appreciated if the authors could validate some of these major genes in primary cells (e.g., activated T lymphocytes or isolated ILC1s), but this would be outside of the scope of the current manuscript. Therefore, no major revisions are necessary in the manuscript's current state.

Author Response

We thank the reviewer for these kind words. We thank the reviewer for their suggestion and agree that future studies are necessary to validate these findings in primary cells. To address this aspect, we modified one statement in the discussion section as follows: “Further studies will validate these findings in primary cells and help to unravel the role of these cells in the local inflammatory environment, as well as their therapeutic vulnerability towards existing and novel treatments”.

Reviewer 2 Report

Put your hypothesis clearly upfront; otherwise your scholarship becomes more a retrospective conclusion paper..

The data has a problem. By combining data derived from other inflammatory diseases of unknown pathogenesis with that of Crohn's disease which has a well-defined pathogenesis, you have unintentionally compromised the significance, but probably not the validity of the findings. Genetic susceptibility is a prime determinant of disease vs infection.

You collectively did a nice job with presenting the data. Next publication focus on explaining the implied WHYs and HOWs of your data in the discussion to avert being labelled a measurement paper. 

Author Response

We thank the reviewer for these helpful suggestions.

To address the reviewer’s concerns, we elaborated on our hypotheses we had before initiating this project. The revised introduction contains the following paragraph: “We hypothesized that only specific cell types mediate the induction of these genes. Specifically, due to their key role in maintaining the intestinal barrier function and their presumably high abundance in samples utilized for bulk sequencing datasets, we expect that epithelial cells largely contribute to the induction of CD-risk genes. In addition, we hypothesize that distinct immune regulatory cells play major roles in mediating inflammatory signaling.”.

We absolutely agree that combining data of two IBD entities, CD and UC, in the same study can be problematic. Due to the lack of comprehensive UC single cell datasets, we decided to mainly focus on CD. Due to the high overlap between genes induced in CD and UC, we then decided to investigate the regulation of CD risk genes in UC as well as UC-associated neoplasia and, finally, CRC. Our studies demonstrate that a high proportion of CD-induced genes appears to be deregulated in IBD in general as well as CRC. To clearly state the advantages and disadvantages of the combination of CD with UC within the same study, we added following paragraph to the discussion section “The analysis of CD and UC data within the same study may have reduced the pathogenic insights into the individual respective disease. However, the large proportion of CD-induced genes upregulated in UC suggests a general deregulation of distinct genes as well as signaling pathways in IBD. Future research will aim to specifically verify and expand our findings in CD and UC in separate studies.”. To build upon our findings, we will further focus on the mechanisms underlying the development of inflammatory bowel diseases as well as colorectal cancer in our future studies.